# Evaluation of Soil-Cutting and Plant-Crushing Performance of Rotary Blades with Double-Eccentric Circular-Edge Curve for Harvesting *Cyperus esculentus*

**Hao Zhu, Xiaoning He, Shuqi Shang, Zhuang Zhao, Haiqing Wang, Ying Tan, Chengpeng Li and Dongwei Wang \***

College of Mechanical and Electrical Engineering, Qingdao Agricultural University, Qingdao 266109, China; 20202204029@stu.qau.edu.cn (H.Z.); 201502004@qau.edu.cn (X.H.); sqshang@qau.edu.cn (S.S.); 20192204158@stu.qau.edu.cn (Z.Z.); 20202204018@stu.qau.edu.cn (H.W.); 20202204017@stu.qau.edu.cn (Y.T.); 20202204011@stu.qau.edu.cn (C.L.)
\* Correspondence: 200701031@qau.edu.cn

**Abstract:** Severe plant entanglement and high power consumption are the main problems of the up-cut rotary blade during *Cyperus esculentus* harvesting. Optimization of the rotary blade edge can enhance the soil-cutting and plant-crushing performance. In this study, the double-eccentric circle method was used to design the edge curve of the IT245 rotary blade. The edge curve's dynamic sliding-cutting angle of equidistant points was analyzed to verify that the optimized rotary blade (IT245P) met the requirements of *Cyperus esculentus* harvesting. In order to accurately simulate the fragmentation of *Cyperus esculentus* plants after interaction with the rotary blade, the Hertz–Mindlin with Bonding contact model was selected to establish the flexible model of *Cyperus esculentus* plants. The plant–soil–rotary blade discrete element model was constructed to conduct simulation tests with power consumption and the plant-crushing ratio as evaluation indexes. The field experiment was carried out with tillage depth stability and power consumption as the experimental indexes. The results of the simulation test and field experiment showed that the power consumption of the IT245P rotary blade was reduced by 13.10%, and the plant-crushing rate was increased by 11.75% compared with the IT245 rotary blade. The optimal operating parameters were 1.08 m/s for forward speed, 107.11 mm for tillage depth, and 258.05 r/min for shaft speed. Under such a combination, the tillage depth stability and the power consumption were 94.63% and 42.35 kW. This study showed that a rotary blade with a double-eccentric circular curve could better realize plant-crushing and consumption reduction and meet the operation requirements of *Cyperus esculentus* and other Chinese medicinal materials' harvesting.

**Keywords:** up-cut rotary blade; edge curve; *Cyperus esculentus* plant; flexible model

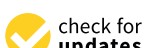

## 1. Introduction

*Cyperus esculentus* is an excellent oilseed crop, originating in Africa and the Mediterranean region, and is known as the "underground walnut". As a high-end healthy edible oil, it is popular among the public [1–3]. *Cyperus esculentus* is still harvested in the semi-mechanized method in most areas in China, in which the excavation job is mostly carried out using the IT245 rotary blade. However, due to the strong tillering ability and developed root system of *Cyperus esculentus*, the plant-crushing performance of the IT245 rotary blade is poor, which affects the subsequent cleaning performance [4–6]. In addition, the root of the rotary blade is particularly easily entangled by plants, which increases the power consumption and seriously hinders the development of the mechanized harvesting of *Cyperus esculentus* [7–9].

The plant-crushing performance is mainly related to the rotary blade's structural parameters, where the rotary blade edge curve is the influencing factor of the plant-crushing performance [10–13]. Similar to *Pinellia ternata* and other traditional Chinese medicine

crops, *Cyperus esculentus* has high requirements for plant-crushing. Scholars at home and abroad have conducted numerous researches on the optimal design of rotary blade edge curves. Depending on the various cutting requirements, the edge curve needs to be designed with different dynamic slip-cutting angle trends. There are two main trends of the dynamic sliding–cutting angle, one that remains constant and one that increases gradually from the tool holder to the tip. In the first one, the edge curve with an equal sliding–cutting angle is designed by a logarithmic spiral for cutting stalks of rice [14–19], corn [20–22], sunflower [23], and other crops. The edge curve with an equal sliding–cutting angle has better cutting performance and is suitable for cutting crops with coarse stalks independent of each other. Also, serrations can be added to the edge to achieve a better tearing effect [16,24–26]. In the second one, the edge curve with a gradually increasing sliding–cutting angle is designed using the Archimedean spiral and sinusoid curve, mainly used to cut objects with high cutting resistance such as clay [27–32]. The gradually increasing sliding–cutting angle can enhance the cutting performance of the cutting edge and thus reduce the working resistance. When designing the edge curve, some scholars [10,11,13] believe that the starting position of the cutting edge of the rotary blade is not the main cutting area, and the dynamic sliding–cutting angle can be reduced appropriately. However, in practical operations in areas with many plant stems, the tool holder is particularly easily entangled by plants, resulting in increased working resistance and power consumption [33,34].

In order to reduce the tillage resistance and enhance the plant-crushing performance of the rotary blade, a rotary blade edge curve was designed based on the double-eccentric circle method in this paper. Based on the discrete element method, a flexible model of the *Cyperus esculentus* plant was established. The power consumption and plant-crushing performance of IT245 and IT245P rotary blades were compared by analyzing the changes in shaft torque and broken bond numbers during the simulation test. With the evaluation indexes of tillage depth stability and power consumption, the best combination of parameters for *Cyperus esculentus* excavation was determined by field experiments to provide a reference for a reduction in power consumption for harvesting *Cyperus esculentus* and other Chinese medicinal materials as well as the optimization of the reverse rotary tiller.

## 2. Materials and Methods

### 2.1. General Structure and Working Principle

#### 2.1.1. General Structure and Parameters

The reverse rotary tiller is composed of a frame, transmission device, shaft, rotary blade, depth limiting device, and leading flow cover; the general structure is shown in Figure 1. The rotary tiller hangs on the tractor's rear, and a one-time operation can complete the rotary tillage, throwing, and other processes. The main structural parameters are shown in Table 1.

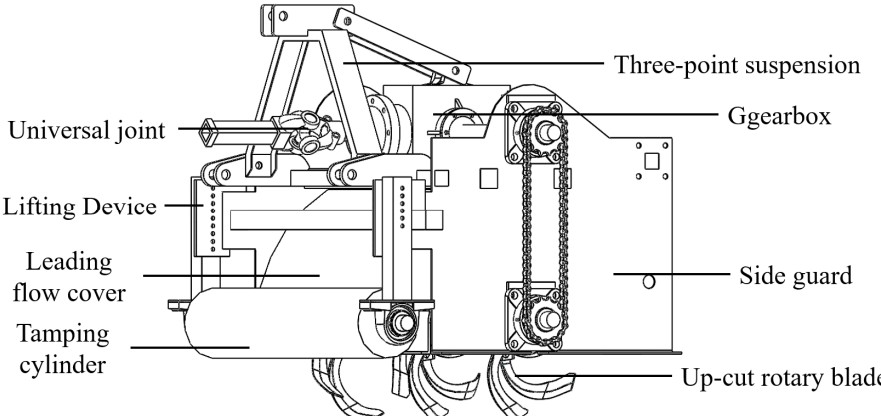

**Figure 1.** General structure of reverse rotary tiller.

**Table 1.** Main parameters of reverse rotary tiller.

| Parameters | Values |
|---|---|
| Matching power/(kW) | 80 |
| Dimension (L × W × H)/(mm) | 1480 × 950 × 880 |
| Operation efficiency/(km·h$^{-1}$) | 0~1.5 |
| Working width/(mm) | 1200 |
| Shaft speed/(r·min$^{-1}$) | 0~400 |
| Tillage depth/(mm) | 0~150 |

The rotary blade is an essential part of the reverse rotary tiller, and its structural parameters have a significant influence on *Cyperus esculentus* harvesting and power consumption. The rotary blade is composed of the lengthwise section, sidelong section, and holding section, as shown in Figure 2. The edge curve of the lengthwise blade has the function of cutting the soil, plant stems, and stubble; the front of the scoop surface has the function of cutting, turning, braking, and throwing the soil.

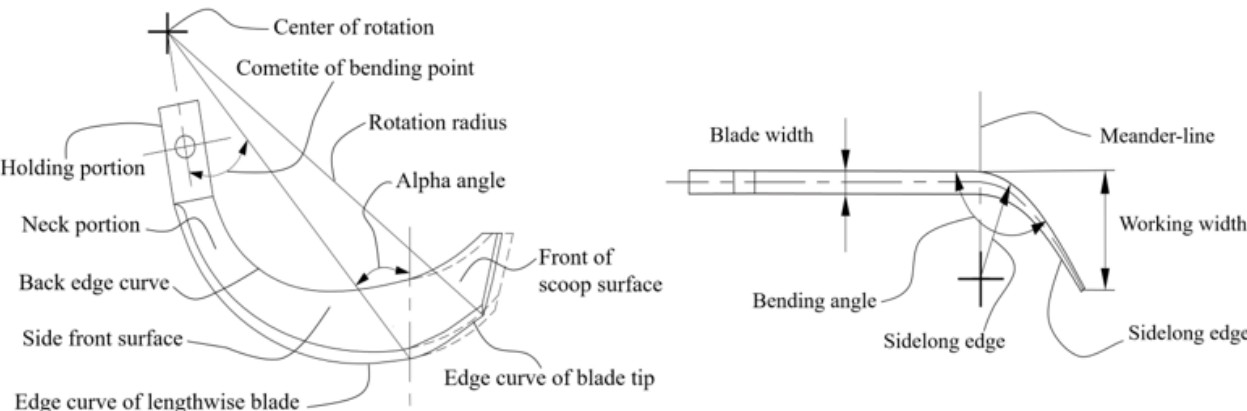

**Figure 2.** Structure of rotary blade.

2.1.2. Working Principle

During field operation of the reverse rotary tiller, the device is connected to the tractor by three-point suspension. The lifting device is adjusted to control the height of the tamping cylinder so that the rotary tillage depth reaches the *Cyperus esculentus* harvesting operation requirements. The tractor output shaft is connected to the transmission input shaft through the universal joint. The rotary tiller shaft is driven by the transmission output shaft to rotate and cut the soil by chain transmission. The rotation direction of the shaft is opposite to that of the tractor's driving wheel. Under the reverse rotation, the bean–soil mixture is thrown backward along the cover to the conveying device to complete the *Cyperus esculentus* harvesting and excavation operation [35].

*2.2. Design of Rotary Blade Edge Curve*

The Archimedean spiral is often adopted to design the edge curve of Chinese standard rotary blades. However, the small sliding–cutting angle at the tool holder makes it easy to tangle plants and increases the excavating resistance and power consumption when working on the land with many plant stems. In this study, based on the IT245 rotary blade, two eccentric arcs with different sliding–cutting performances were used as the edge curve, which can better meet the requirements of *Cyperus esculentus* harvesting. The polar coordinate equation of the eccentric circle was calculated by

$$r = \rho \cos \varphi \pm \rho \sqrt{(R/\rho)^2 - \sin^2 \varphi} \tag{1}$$

where $r$ is the polar radius, mm; $\rho$ is the distance between $O$ and $O_1$, mm; $\varphi$ is the angle between the polar radius of any point on the circle and the $x$-axis; $R$ is the radius of the eccentric circle, mm. The derivative function of Equation (1) was

$$\frac{dr}{d\varphi} = -\rho \sin \varphi + \rho \frac{-\sin \varphi \cos \varphi}{\sqrt{e^2 - \sin^2 \varphi}} \tag{2}$$

where $e$ is the coefficient of excentralization. The static sliding–cutting angle of the eccentric circle $\tau_s$ was calculated by

$$\tau_s = \tan^{-1}\left(\frac{\sqrt{e^2 - \sin^2 \varphi}}{\sin \varphi}\right) \tag{3}$$

As shown in Equation (3), the sliding–cutting angle of the eccentric circle edge curve is related to the coefficient of excentralization and the angle between the pole radius and $x$-axis. The variation law is shown in Figure 3. When the sliding–cutting angle is small, the high cutting resistance is not conducive to sliding–cutting, and it is easily entangled by plants, causing an increase in cutting load. When the sliding–cutting angle increases, the cutting resistance will be reduced, conducive to sliding–cutting, and can effectively prevent plant entanglement. However, the cutting distance increases, and the frictional resistance continues to work, resulting in increased power consumption. According to the research results [18,24], the optimal range of sliding–cutting angles is 35°~55°. In addition, for smooth sliding of uncut plant stems out of the blade during multi-plant stem field operations, the sliding–cutting angle of the rotary blade is required to increase gradually from the beginning to the end. Referring to the variation law of eccentric circle $\tau$-$\varphi$, the coefficient of excentralization was taken as 1.2~1.6 to meet the sliding–cutting angle requirements for *Cyperus esculentus* harvesting.

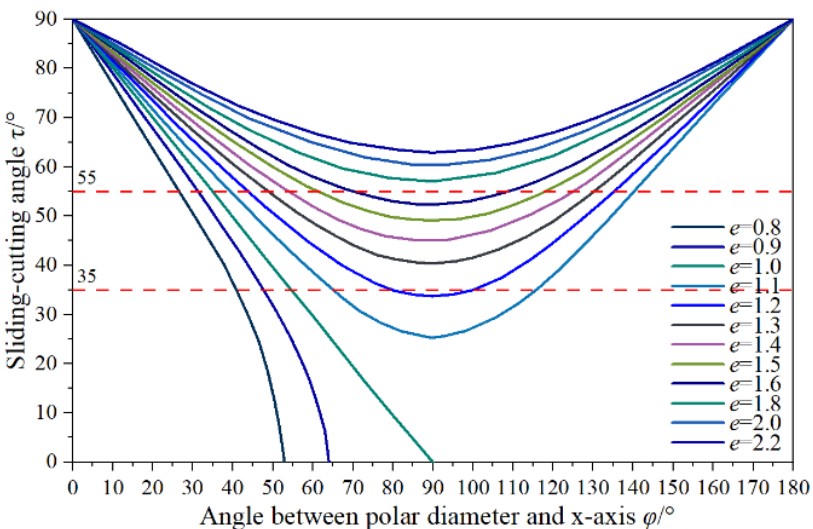

**Figure 3.** Variation law between sliding–cutting angle and angle between polar radius and *x*-axis.

In order to reduce the plant entanglement at the root of the rotary blade, the sliding–cutting angle needs to meet the conditions of uncut plant stems sliding out. Since the tool holder is the secondary working area, the blade length in this area can be reduced, and the sliding–cutting angle should be increased quickly. When the plant stems move to the middle of the cutting edge, the sliding–cutting angle should increase slowly with the increase of the pole radius. The plant stems are cut off or slide out of the rotary tillage area along the cutting edge due to the pushing action. According to the requirements of *Cyperus esculentus* harvesting and the variation law of eccentric circle τc-φ, the coefficient of excentralization of sections *AB* and *BC* was 1.2 and 1.6. In order to prevent the tool holder

from plant entanglement, the minimum working radius $r_0$ was reduced to 125 mm within a reasonable range. The extreme diameter $r_1$ at the bending line was 230 mm, consistent with the tillage depth requirements of *Cyperus esculentus* harvesting. The coordinate system was established with the rotation center as the origin, and the double-eccentric circle curves *AB* and *BC* are shown in Figure 4. The eccentric circle radius *R* and the polar angle $\varphi$ at any point on the eccentric circle curve were calculated by:

$$R = -\frac{re}{\cos\varphi + \sqrt{e^2 - \sin^2\varphi}} \tag{4}$$

$$\cos\varphi = \frac{r^2 - (R/e)^2(e^2 - 1)}{2rR/e} = \frac{e^2 r^2 - R^2(e^2 - 1)}{2rRe} \tag{5}$$

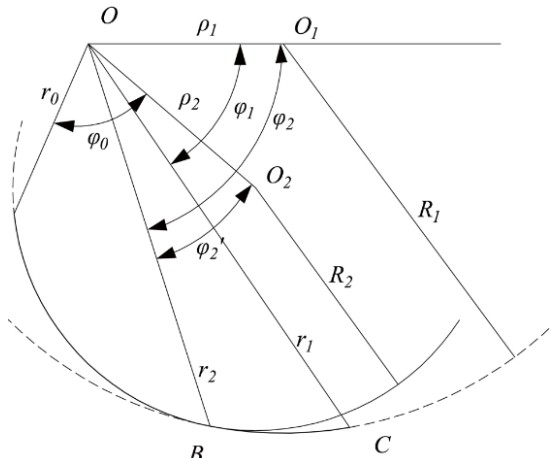

**Figure 4.** Double-eccentric circle curve, where $O_1$ and $O_2$ are the centers of the eccentric circle; $\varphi_0$ and $\varphi'_2$ are polar angles of points *A* and *B* on curve *AB*; $\varphi_1$ and $\varphi_2$ are polar angles of points *C* and *B* on curve *BC*; $r_0$, $r_1$, and $r_2$ are polar radii of point *A, C,* and *B*; $R_1$ and $R_2$ are the radii of the eccentric circle; $\rho_1$ is the distance between *O* and $O_1$; $\rho_2$ is the distance between *O* and $O_2$.

The double-eccentric circular-edge design should ensure a smooth transition at the connection of the two arcs. Point *B* is the intersection point of the two arcs, in which the sliding–cutting angle of the two arcs should be equal:

$$\operatorname{tg}\tau_s = \frac{\sqrt{e_2^2 - \sin^2\varphi'_2}}{\sin\varphi'_2} = \frac{\sqrt{e_1^2 - \sin^2\varphi_2}}{\sin\varphi_2} \tag{6}$$

The polar angle $\varphi'_2$ at point *B* on the eccentric arc *AB* was calculated by

$$\varphi'_2 = \sin^{-1}\frac{e_2}{\sqrt{1 + (e_1^2 - \sin^2\varphi_2)/(\sin^2\varphi_2)}} \tag{7}$$

The radius $R_2$ of the eccentric circle and the polar angle $\varphi_0$ at point *A* were calculated by Equations (6) and (7). In the field operation of the rotary tiller, there is an angle $\Delta\tau$ between the absolute velocity $v_a$ and the relative velocity $v_r$ at each point on the edge curve. $\Delta\tau$ is the difference between the dynamic sliding–cutting angle and the static sliding–cutting angle of the up-cut rotary blade, as shown in Figure 5. According to the trigonometric function relationship, $\Delta\tau$ was calculated by

$$\begin{cases} v_a^2 = r^2\omega^2 + v_e^2 + 2r\omega v_e \cos\varphi \\ v_e^2 = v_a^2 + r^2\omega^2 - 2v_a r\omega \cos\Delta\tau \end{cases} \tag{8}$$

$$\Delta\tau = \arctan(v_e / r\omega) \tag{9}$$

where $v_e$ is the forward speed, m/s. The suitable operation parameters of the rotary tiller are 0.8~1.4 km/h for forward speed and 250~350 r/min for shaft speed [35,36]. According to Equation (9), $\Delta\tau$ is proportional to the forward speed $v_e$ and inversely proportional to the shaft speed $\omega$. Therefore, the operating conditions of 1.4 km/h for forward speed and 250 r/min for shaft speed were chosen as the critical conditions for the edge-curve design [37,38]. It ensured that the rotary blade met the sliding–cutting conditions under low shaft speed and high forward speed.

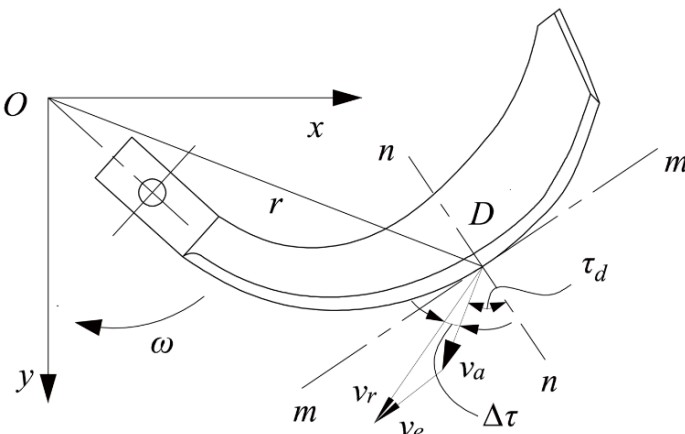

**Figure 5.** Speed analysis of cutting edge of rotary blade, where *D* is an arbitrary point on the rotary blade cutting edge; *n-n* is the normal line at point *D*; *m-m* is the tangent line at point *D*.

After all the double-eccentric circle cutting-edge parameters were determined, four points were selected proportionally on each of the eccentric arcs *AB* and *BC* to solve the dynamic sliding–cutting angle. The results are shown in Table 2.

**Table 2.** Calculated values of parameters of edge curve of lengthwise blade.

| Eccentric Circular Arc | NO. | Polar Radius $r$/mm | Polar Angle/(°) | Static Cutting Angle $\varphi$/(°) | Static Sliding–Cutting Angle $\tau_s$/(°) | Dynamic Sliding–Cutting Angle $\tau_d$/(°) | Difference between $\tau_s$ and $\tau_d$ $\Delta\tau$/(°) |
|---|---|---|---|---|---|---|---|
| *AB* | 1 | 125.00 | 63.26 | 48.07 | 41.93 | 35.15 | 6.78 |
| | 2 | 137.34 | 58.16 | 45.05 | 44.95 | 38.77 | 6.18 |
| | 3 | 149.48 | 53.03 | 41.73 | 48.27 | 42.59 | 5.68 |
| | 4 | 160.00 | 48.42 | 38.54 | 51.46 | 46.15 | 5.31 |
| *BC* | 1 | 174.39 | 80.22 | 38.01 | 51.99 | 47.11 | 4.87 |
| | 2 | 196.35 | 71.28 | 36.29 | 53.71 | 49.38 | 4.33 |
| | 3 | 214.73 | 64.04 | 34.18 | 55.82 | 51.86 | 3.96 |
| | 4 | 230.00 | 58.01 | 32.37 | 58.00 | 54.30 | 3.70 |

As shown in Table 2, the difference between the static sliding–cutting angle and dynamic sliding–cutting angle was inversely proportional to the pole radius. The dynamic sliding–cutting angle of the cutting edge was within 35°~55°. In the eccentric arc (*AB*) at the rotary blade tool holder, the dynamic sliding–cutting angle increased rapidly from 35.15° to 46.15° with the increase in the pole diameter. The eccentric arc (*AB*) was short as the edge curve of the secondary operation area. The dynamic sliding–cutting angle at the starting point was greater than 35° under extreme operational conditions, which can theoretically solve the problem of plant entanglement at the rotary blade tool holder and meet the preset optimization target. In the eccentric arc (*BC*), the dynamic sliding–cutting angle increased slowly from 46.15° to 54.30° with the increase in the pole radius. The eccentric arc (*BC*) was

long as the edge curve of the main operation area. The dynamic sliding–cutting angle was large, with good sliding–cutting performance for cutting soil and crushing plants.

The front of the scoop surface is an essential part of the rotary blade, which has the function of turning, braking, and throwing soil during the rotary tillage operation. In order to ensure sliding–cutting performance and smooth transition and prevent plant entanglement caused by a sudden change in the cutting-edge curve curvature, the front cutting edge was designed with the same eccentric circle curve as the end of the side cutting edge.

## *2.3. Discrete Element Simulation*
### 2.3.1. Rotary Blade Model

SolidWorks 2020 (Dassault Systemes S.A) was applied to build a simplified model of the up-cut rotary blade on a 1:1 scale. The model was saved in STL format and imported into EDEM2021 (DEM Solutions Ltd., Edinburgh, Scotland, UK). The simulation model is shown in Figure 6. The material of both rotary blades was 65Mn steel [21].

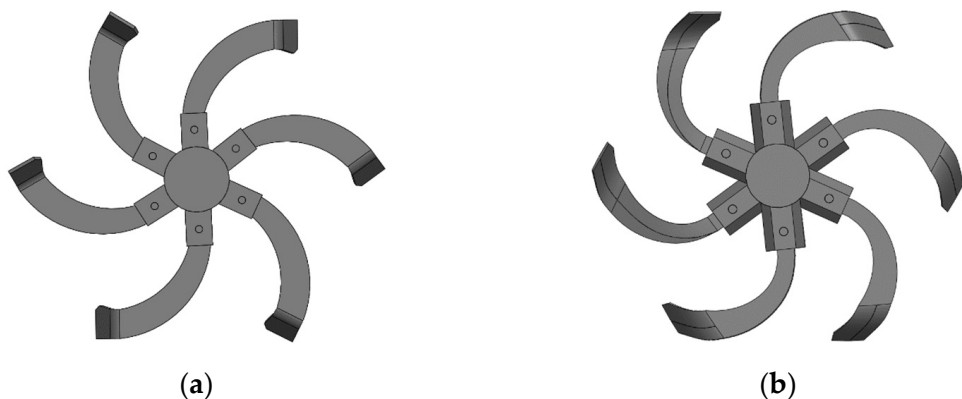

|  (**a**)  |  (**b**)  |

**Figure 6.** Simulation model of up-cut rotary blade. (**a**) IT245 rotary blade; (**b**) IT245P rotary blade.

### 2.3.2. Soil Model

The soils in this study were sampled by the five-point sampling method at the *Cyperus esculentus* plantation in Minquan County, Shangqiu City, Henan Province (115°18′ E, 34°31′ N), China, in mid-November 2020. The soil's water content, density, and firmness at different depths were measured. The experimental instruments were an RXH-14B hot-air circulation oven (Nanjing Changjiang Pharmaceutical Machinery Manufacturing Ltd., Nanjing, China), JDM-I automatic soil-density tester (Inner Mongolia Huake Jiacheng Technology Ltd., Ordos, China), TJSD-750II soil-compactness tester (Zhejiang Topunnong Technology Ltd., Hangzhou, China), soil-extraction ring knife, and shovel.

The SMJYS200 soil-testing vibrating sieve (Henan Shuangmu Machinery Manufacturing Ltd., Xinxiang, China) was used for sieving. The soil type of the *Cyperus esculentus* planting base was sandy loam, in which the clay particles (<0.002 mm) account for 10.34 ± 1.64%, powder sand (0.02~0.002 mm) accounts for 22.48 ± 1.94%, and sand particles (0.02~2 mm) account for 67.18 ± 3.95%. Due to the minimal diameter of sandy loam particles, the number of soil particles in the simulation test reached ten million, which requires high computer performance and a long simulation time. The particle scaling method is a common and more feasible processing method to ensure test accuracy. The particles are appropriately scaled up so that the discrete element model can complete the simulation within a reasonable and adequate time [16,18]. In this study, the radius of a single soil particle was set to 2 mm according to the reference literature [17,20,21] and soil parameters of the *Cyperus esculentus* planting site. The following three models were set according to the structure of sandy loam soil particles: granular particles, columnar particles, and agglomerated particles, as shown in Figure 7.

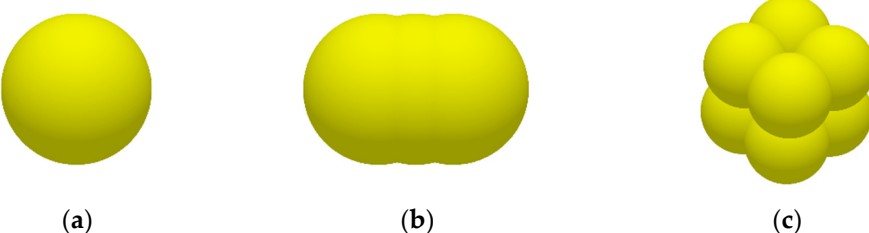

(a)　　　　　　　　(b)　　　　　　　　(c)

**Figure 7.** Sandy loam soil particle model. (**a**) Granular particle; (**b**) Columnar particle; (**c**) Agglomerated particle.

According to the intrinsic parameters of sandy loam soil, the soil state is loose and the cohesion between soil particles is low. The Hertz–Mindlin (no-slip) contact model does not consider the mutual attraction between particles and is suitable for studying sandy loam soil conditions, as shown in Figure 8. Therefore, the Hertz–Mindlin (no-slip) contact model was chosen as the contact mechanics model for the discrete element simulation in this study.

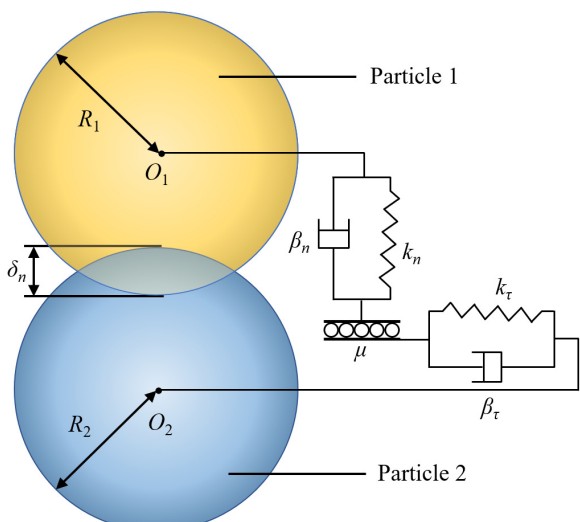

**Figure 8.** Hertz–Mindlin (no-slip) contact model. where $O_1$ and $O_2$ are the positions of the sphere centers of two particles, respectively; $R_1$ and $R_2$ are the radii of two particles, respectively, mm; $\delta_n$ is normal overlap after collisions, mm; $k_n$ and $k_\tau$ are the normal and tangential Hooke coefficients of particles, respectively; $\beta_n$ and $\beta_\tau$ are the normal and tangential damping coefficients of particles, respectively; $\mu$ is the sliding coefficient between particles.

### 2.3.3. *Cyperus esculentus* Plant Model

The object of this study was Henan No.1 *Cyperus esculentus*, which was sampled by a random sampling method, as shown in Figure 9. The average length, width, and thickness of plant leaves were 82.35 $\pm$ 5.74 mm, 5.12 $\pm$ 0.62 mm, and 1.52 $\pm$ 0.36 mm. The average length and diameter of the root system were 135.46 $\pm$ 9.87 mm and 0.84 $\pm$ 0.27 mm. The average moisture content of the leaves and root system of the *Cyperus esculentus* plants measured by the RXH-14B hot-air circulation oven was 24.25 $\pm$ 4.90% and 37.60 $\pm$ 5.44%.

The characterization of *Cyperus esculentus* plants is complex, and the morphological changes are diverse after stress. In order to simulate the interparticle cohesion accurately, the Hertz–Mindlin with Bonding contact model was chosen to constrain the plant morphology and simulate the fragmentation of the leaves and roots of the *Cyperus esculentus* plant during rotary tillage. After the constraint was broken, bonds were no longer generated between the two particles and the contact was solved as a separate sphere model. In this soil bonding model, when the practical contact distance between soil particles was less

than the sum of the bond radius ($R_{1B} + R_{2B}$), the constraint was formed by the adhesive bond between particles [39,40], as shown in Figure 10.

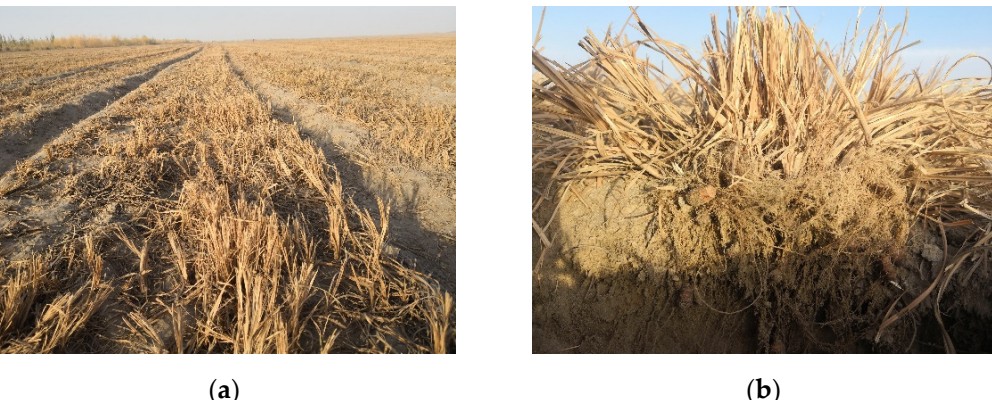

(**a**)                    (**b**)

**Figure 9.** *Cyperus esculentus* planting base. (**a**) Uncultivated land; (**b**) section of the *Cyperus esculentus* plant.

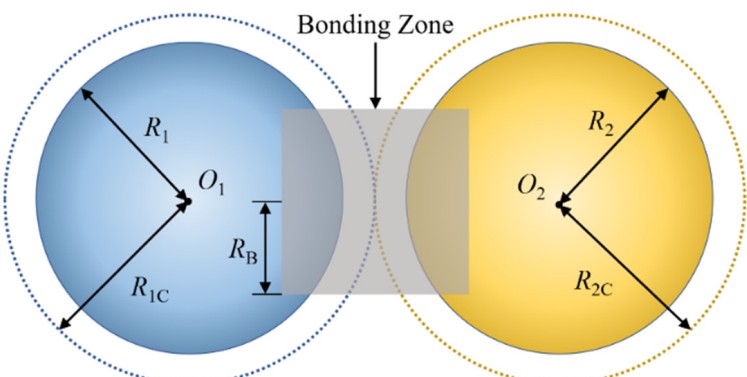

**Figure 10.** Hertz–Mindlin with Bonding contact model, where $R_1$ and $R_2$ are the practical radii of particles, mm; $R_{1C}$ and $R_{2C}$ are the contact radii between particles, mm; $R_B$ is the bonding zone radius between particles, mm.

In a specific volume of material, the bonding zone formed between particles can be transformed into a bonding force due to the presence of water. Based on the *Cyperus esculentus* plant parameters determination results, the root particles were simplified to spheres of the same diameter and the leaf particles were simplified to two spheres connected, as shown in Figure 11. The bonding radius was calculated by

$$\omega = \frac{m_2}{m_1 + m_2} = \frac{\rho_2\left(\pi R_B^3 - \pi R^3\right)}{\rho_1 \pi R^3 + \rho_2\left(\pi R_B^3 - \pi R^3\right)} \tag{10}$$

where $\omega$ is the moisture content,%; $m_1$ is the mass of the particle, kg; $m_2$ is the mass of moisture, kg; $\rho_1$ is the dry density of the material particle, kg/m³; $\rho_2$ is the density of moisture, kg/m³; $R$ is the material particle radius, m; $R_B$ is the bonding radius between particles, m.

The bonding radius of *Cyperus esculentus* leaf and root particles was 3.35 mm and 2.25 mm. A new meta-particle was created, and 142 leaf particles and 401 root particles were added according to the morphology and dimensions of *Cyperus esculentus* plants. The triaxial coordinates were set separately to establish a discrete metamodel of *Cyperus esculentus* plants, as shown in Figure 11 [41–44]. Contact parameters of discrete element model are shown in Table 3.

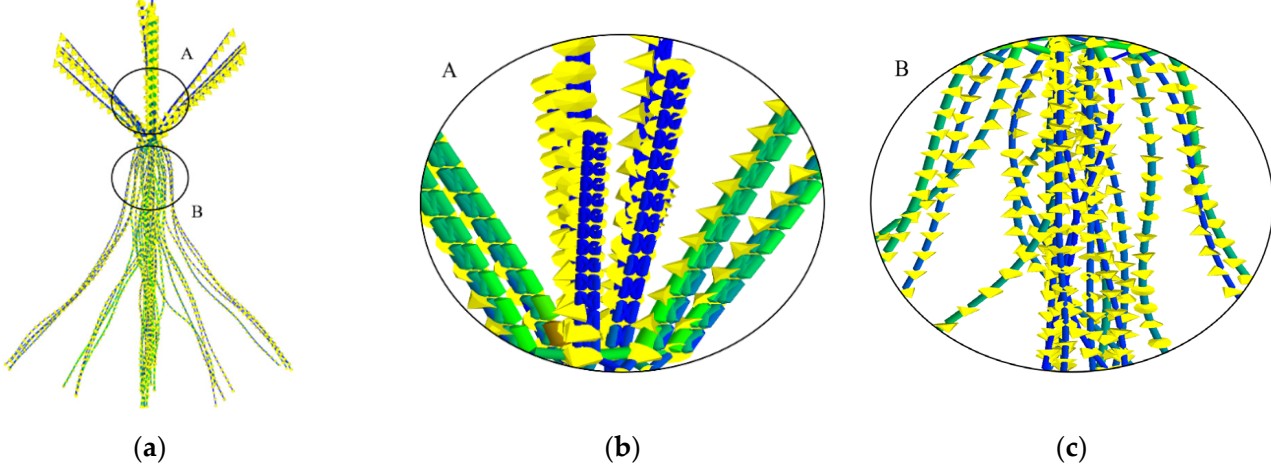

**Figure 11.** Flexible model of *Cyperus esculentus* plants. (**a**) Whole plant; (**b**) bonds between leaf particles; (**c**) bonds between root particles.

**Table 3.** Contact parameters of discrete element model.

| Contact Model | Static Friction Coefficient | Rolling Friction Coefficient | Restitution Coefficient |
|---|---|---|---|
| Soil–Soil | 0.55 | 0.15 | 0.43 |
| Plant–Plant | 0.34 | 0.08 | 0.35 |
| Soil–Plant | 0.48 | 0.05 | 0.32 |
| Soil–65Mn | 0.52 | 0.12 | 0.20 |
| Plant–65Mn | 0.33 | 0.10 | 0.30 |

2.3.4. Discrete Element Simulation of Soil–Plant–Rotary Blade

Based on the trial soil slot of Qingdao Agricultural University, the simulated trial soil slot was established with a length × width × height of 1000 mm × 550 mm × 360 mm and the material was 45 steel. According to the *Cyperus esculentus* planting pattern, eight plants with a length × width × height of 280 mm × 200 mm × 200 mm were established, and the plant spacing and row spacing were 200 mm and 300 mm, respectively. After the rotary blade model was imported into EDEM, the "Sinusoidal Translation Kinematic" and "Sinusoidal Rotationnematic Kinematic" motions were added: 0.8 m/s for the forward speed and 250 r/min for the shaft speed. The simulation process of reverse rotary tillage for *Cyperus esculentus* harvesting is shown in Figure 12. At 0~0.5 s, *Cyperus esculentus* plant model was generated; 0.5~1 s, soil particle model was generated statically; 1~9 s, simulated rotary tillage operation was conducted.

*2.4. Evaluation Index of Rotary Tillage Quality*

2.4.1. Power Consumption

During the simulation of the rotary tillage, the difference in power consumption can be reflected by the dynamic changes in shaft torque and forward resistance. The shaft speed and forward speed of the rotary blade were 250 r/min and 0.5 m/s, respectively. The power consumption of the rotary tiller was calculated by

$$W = \frac{nM}{9550} + Fv_m \tag{11}$$

where $W$ is the power consumption, kW; $n$ is the shaft speed, r/min; $M$ is the shaft torque, N·m; $F$ is the forward resistance, N; $v_m$ is the forward speed, m/s.

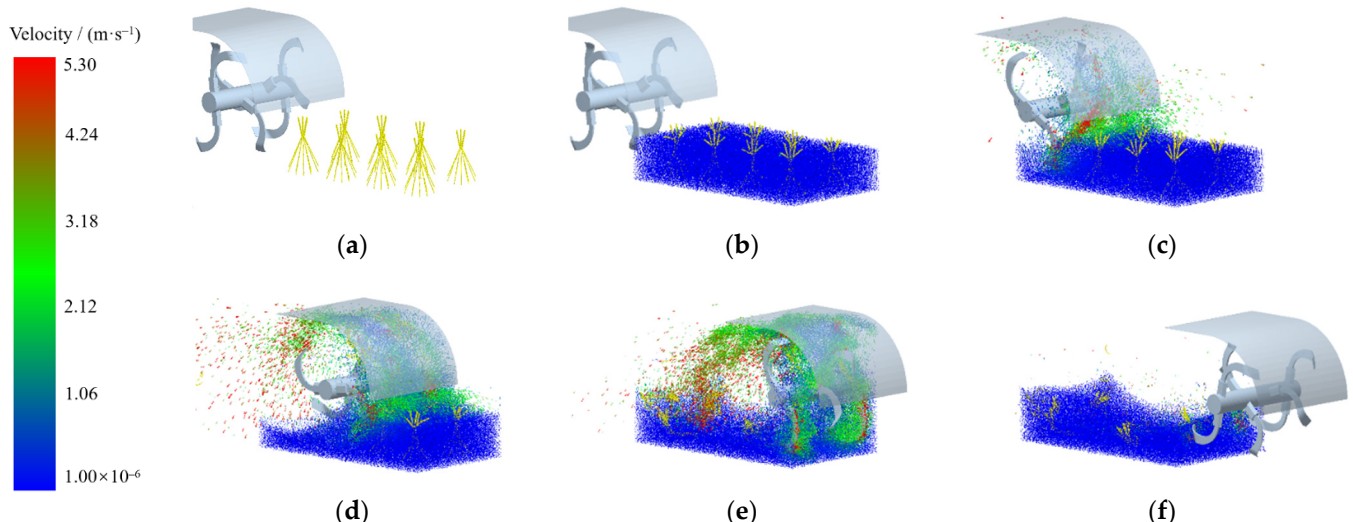

**Figure 12.** Simulation process of reverse rotary tillage for *Cyperus esculentus* harvesting. (**a**) 0.5 s; (**b**) 1 s; (**c**) 2 s; (**d**) 4.5 s; (**e**) 6.4 s; (**f**) 8.7 s.

2.4.2. Plant-Crushing Performance

In the initial state, the total number of intact bonds and broken bonds of the *Cyperus esculentus* plant model were 5560 and 0. The soil particles and the *Cyperus esculentus* plant particles were hidden in the reverse-rotary-tillage simulation. Only the intact bonds between the *Cyperus esculentus* plant particles were retained. With the rotary blade cutting, the *Cyperus esculentus* plant model reached the force limit, and the bonding zone was broken [45–47]. The process of breaking the bonding zone of *Cyperus esculentus* plant particles is shown in Figure 13.

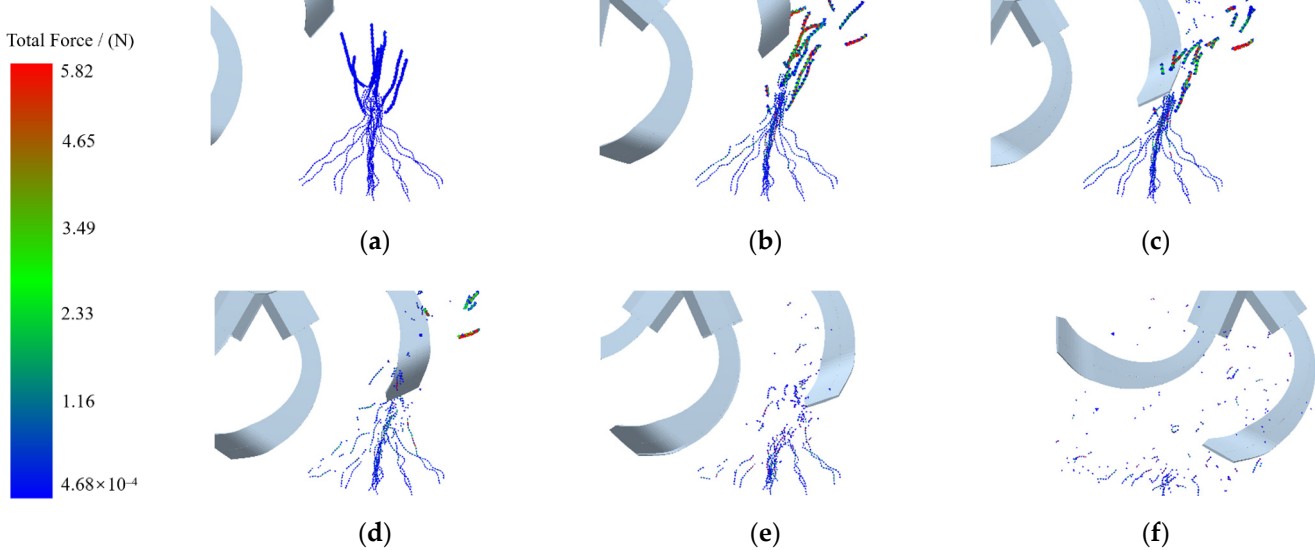

**Figure 13.** The crushing process of *Cyperus esculentus* plants. (**a**) 1.9 s; (**b**) 2.1 s; (**c**) 2.3 s; (**d**) 2.5 s; (**e**) 2.7 s; (**f**) 2.9 s.

*2.5. Field Experiment*

2.5.1. Test Conditions

To test the field operation performance of the reverse rotary tiller with an optimized edge curve, a field experiment of reverse rotary tillage excavation was conducted on 19 November 2021 at the *Cyperus esculentus* planting base in Minquan County, Shangqiu City, Henan Province (115°18′ E, 34°31′ N), as shown in Figure 14. Based on the GB/T

5668-2017 "Rotary Tiller", a representative field of *Cyperus esculentus* was selected and the test conditions around the field were ensured to be the same [43,47,48]. The length of the test area was 50 m, a 10 m stabilization area was set at both sides, and the width was 32 m. Four routes were tested during the field experiment, with an 8 m interval between adjacent routes. The test devices were the Dongfeng DF604 tractor (Changzhou Dongfeng Agricultural Machinery Group Ltd., Changzhou, China, matching power: 60 hp) and the reverse-rotary-tillage device.

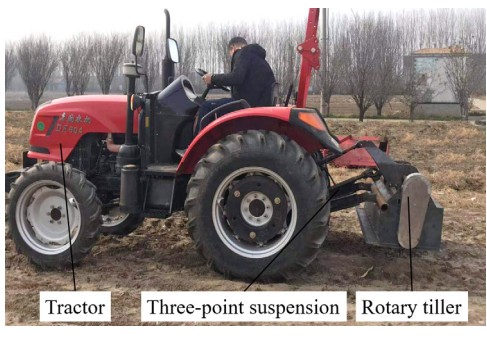
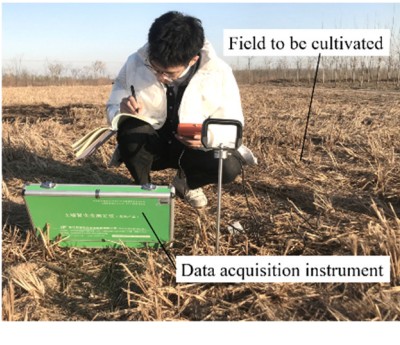

(**a**)　　　　　　　　　　　　　　　　　　　　　(**b**)

**Figure 14.** Field experiment. (**a**) Test device; (**b**) data acquisition.

The environmental parameters of the field experiment were measured in the tillage area and the results are shown in Table 4. The data collection instruments were a RXH-14-B hot-air circulation oven (Nanjing Changjiang Pharmaceutical Machinery Manufacturing Ltd.), JDM-I automatic soil-density tester (Inner Mongolia Huake Jiacheng Technology Ltd.), and TJSD-750-II soil-compactness tachometer (Zhejiang Topunnong Technology Ltd., range: 0~7922 kPa, depth: 0~375 mm, accuracy: ±5%). Other auxiliary tools were MPL13-50 steel-frame tape measure (Hunan Maojun Baogong Electronics Ltd., Changsha, China, range: 0~150 m, accuracy: 1 mm), stainless steel straightedge (range: 0~200 mm, accuracy: 0.5 mm), XL-008-type stopwatch (Wenzhou Shengmei Instrumentation Ltd., Wenzhou, China, accuracy: 0.01 s), soil-taking ring knife, and shovel.

**Table 4.** Test conditions at field experiment site.

| Test Site | Soil Type | Soil Moisture Content/(%) | Soil Density /(kg m$^{-3}$) | Soil Firmness/(kPa) | Row Spacing/(mm) | Plant Spacing/(mm) | Stubble Height/(mm) | Yield Per Mu/(kg) |
|---|---|---|---|---|---|---|---|---|
| Minquan County | Sandy Loam | $15.36 \pm 1.56$ | $2650 \pm 50$ | $425 \pm 50$ | $200 \pm 15$ | $150 \pm 10$ | $90 \pm 7$ | $1265 \pm 50$ |

2.5.2. Index and Factor of Test

According to the tillage requirements, when operating under the same working conditions, agricultural equipment should ensure the same tillage depth. The fluctuation of tillage depth will lead to problems such as increased power consumption or an increased missed digging rate. The tillage depth stability was calculated by

$$U = \left(1 - \frac{\sqrt{\sum_{i=1}^{N}(a_i - a)^2/N}}{\sum_{i=1}^{N} a_i/N}\right) \times 100\%$$ (12)

where $U$ is the tillage depth stability coefficient; $a$ is the average tillage depth, mm; $a_i$ is the tillage depth of point $i$, mm; $N$ is the number of measurement points.

The power consumption is an important index to assess the operating quality of the rotary tiller. In the field test, the working width $B_0$ of each test route was measured with a

tape measure; the tillage depth $h_n$ was measured as above; the time used for each test route was measured with a stopwatch to calculate the single-route machine forward speed $v_m$. Concerning the Agricultural Machinery Design Manual [46], the power consumption of the rotary tiller was calculated by

$$W = \frac{1}{\eta} B_0 \frac{h_n}{100} v_m \left( p + \frac{\rho(\pi R n)^2}{7200} \right)$$

(13)

where $W$ is the power consumption, kW; $\eta$ is the transmission efficiency, 0.9 in this study; $B_0$ is the working width, m; $h_n$ is the tillage depth, cm; $v_m$ is the forward speed, m/s; $p$ is the specific resistance of cutting soil, N/m$^2$; $\rho$ is the soil density before tillage, kg/m$^3$; $R$ is the rotating radius of the rotary blade, m; $n$ is the shaft speed, r/min.

In this experiment, the main factors affecting the harvesting quality of *Cyperus esculentus* were forward speed, shaft speed, and tillage depth. The experimental factor codes are shown in Table 5. During the experiment, the forward power was provided by the tractor and controlled by adjusting the tractor gear. The bevel gear ratio in the transmission system of the reverse-rotary-tillage device was 1.5. The chain-transmission ratio was 0.85, 1, and 1.15 from low to high, which cooperated to control the shaft speed. The tillage depth was controlled by adjusting the height of the depth-limiting device.

**Table 5.** Experimental factors codes.

| Test Factor | Symbol | Test Level | | |
|---|---|---|---|---|
| | | −1 | 0 | 1 |
| Forward speed/(km·h$^{-1}$) | $A$ | 0.8 | 1.1 | 1.4 |
| Tillage depth/(mm) | $B$ | 100 | 115 | 130 |
| Shaft speed/(r·min$^{-1}$) | $C$ | 250 | 300 | 350 |

## 3. Results and Analysis

### 3.1. Evaluation Index of Rotary Tillage Quality

3.1.1. Analysis of Power Consumption

The curves of the power consumption of the IT245P rotary blade and IT245 rotary blade are shown in Figure 15. During the rotary tillage excavation simulation test, the overall power consumption change trends of the IT245P rotary blade and IT245 rotary blade were similar and divided into four stages. In the first stage, the *Cyperus esculentus* plant and soil particles were generated, the rotary blade did not interact with the soil, and the power consumption of the rotary tiller was 0. In the second stage, the rotary blade rotated in the reverse direction to cut soil. At the moment of initial contact with the soil, the force on the rotary blade increased steeply, which led to a rapid increase in the power consumption of the rotary tiller. The shaft continued to rotate and the power consumption reached the first peak when the front cutting edge touched the soil. With the increase in tillage depth, the power consumption reached the maximum. In the third stage, when the rotary blade continued to rotate beyond the maximum tillage depth, the resistance of the rotary tiller gradually decreased and the power consumption had a valley value. In the fourth stage, as the rotating tiller moved forward, part of the soil collided with the deflector and fell back into the front of the rotary tiller causing congestion. The cutting resistance and the power consumption increased. The power consumption during the overall reverse-rotary-tillage operation changed periodically and coincided with the practical state of the rotary tiller cutting the soil. In the simulation test, the average power consumption of the IT245 and IT245P rotary blades were 43.90 kW and 38.82 kW, with an average power consumption reduction of 13.10%. The results showed that the optimized rotary blade could better complete the excavation of *Cyperus esculentus* with reduced resistance and consumption.

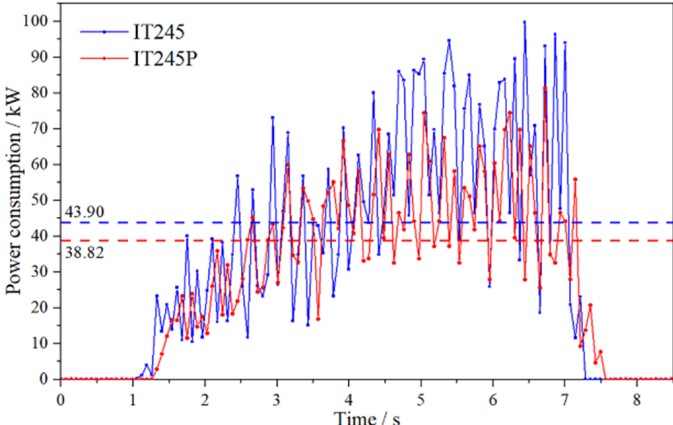

**Figure 15.** Power consumption curve of rotary blade.

3.1.2. Analysis of Plant-Crushing Performance

The curve of the broken bond number of *Cyperus esculentus* plant particles with time is shown in Figure 16. When the IT245 and IT245P rotary blades worked, the number of broken bonds was nonlinearly proportional to time and divided into two stages. At 0~2 s, the rotary blade cut the soil but did not touch the *Cyperus esculentus* plants. The force on the *Cyperus esculentus* plant particles did not reach the breaking limit and it remained in its original state. At 2~9 s, the rotary blade started to cut the *Cyperus esculentus* plant and the force on the plant gradually increased until it reached the limit of force. The bonds between the particles broke, which showed that the *Cyperus esculentus* plants were broken. Furthermore, as the machine moved forward, the number of broken interparticle bonds increased until the rotary tiller left the simulated operation area. The total number of broken interparticle bonds of the IT245 and IT245P rotary blades were 1575 and 1760, and the total number of broken bonds increased by 11.75%. The results showed that the optimized rotary tillage blade was more effective in breaking the *Cyperus esculentus* plants and could reduce the subsequent separation and cleaning-process burden.

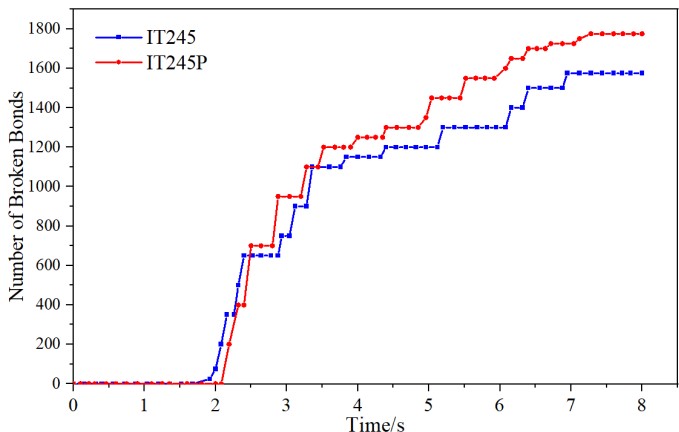

**Figure 16.** Comparison of broken bonds of *Cyperus esculentus* plants.

*3.2. Field Experiment*

According to the Box–Benhnken central combination design theory of the Design-Expert software, the forward speed, shaft speed, and tillage depth of the reverse-rotary-tillage device were selected as the influencing factors for the response surface experimental research, and the response values of tillage depth stability and power consumption were used for parameter optimization by a three-factor, three-level quadratic regression orthogonal test scheme. The results of the orthogonal test are shown in Table 6.

**Table 6.** Experimental protocol and results.

| No. | Test Factor | | | Evaluation Index | |
|---|---|---|---|---|---|
| | Forward Speed | Tillage Depth | Shaft Speed | Tillage Depth Stability $U$/% | Power Consumption $W$/(kW) |
| 1 | 0 | −1 | −1 | 95.99 | 42.51 |
| 2 | −1 | 0 | −1 | 95.01 | 42.03 |
| 3 | −1 | −1 | 0 | 94.63 | 42.20 |
| 4 | 1 | 1 | 0 | 94.78 | 45.90 |
| 5 | −1 | 1 | 0 | 93.21 | 44.35 |
| 6 | 0 | 0 | 0 | 94.94 | 43.43 |
| 7 | 0 | −1 | 1 | 95.64 | 43.67 |
| 8 | 1 | −1 | 0 | 95.96 | 43.84 |
| 9 | 0 | 0 | 0 | 94.74 | 43.41 |
| 10 | 0 | 1 | 1 | 94.47 | 46.59 |
| 11 | 0 | 0 | 0 | 94.89 | 43.17 |
| 12 | 0 | 0 | 0 | 95.05 | 43.01 |
| 13 | 1 | 0 | −1 | 96.04 | 41.84 |
| 14 | 0 | 1 | −1 | 95.02 | 44.87 |
| 15 | 0 | 0 | 0 | 94.82 | 43.82 |
| 16 | −1 | 0 | 1 | 94.79 | 42.90 |
| 17 | 1 | 0 | 1 | 95.74 | 44.63 |

3.2.1. Tillage Depth Stability of Cyperus esculentus Harvesting

Removing non-significant influences, the quadratic polynomial regression equation targeting the tillage depth stability of the rotary tiller was

$$U = 94.89 - 0.59A - 0.18B + 0.61C + 0.57B^2 \qquad (14)$$

From the analysis of the variance of tillage depth stability, as shown in Table 7, the determination coefficient $R^2 = 0.9734$ indicated that the regression equation model applied to 97.34% of the test data. The effect of the regression model was highly significant ($p < 0.01$); the effect of shaft speed ($C$), forward speed ($A$), and the interaction term of tillage depth ($B^2$) on tillage depth stability was highly significant ($p < 0.01$); the effect of tillage depth ($B$) on tillage depth stability was significant ($p < 0.05$). The other test factors' interaction terms did not significantly affect tillage depth stability. Neglecting the non-significant factors and comparing the $F$ values, the influence of the factors on the test mounding angle was $C > A > B$.

**Table 7.** Variance analysis of quadratic term model of tillage depth stability.

| Source | Sum of Squares | Freedom | Mean Square | F Value | p-Value |
|---|---|---|---|---|---|
| Model | 7.54 | 9 | 0.84 | 28.49 | <0.01 ** |
| $A$ | 2.81 | 1 | 2.81 | 95.53 | <0.01 ** |
| $B$ | 0.25 | 1 | 0.25 | 8.57 | 0.0221 * |
| $C$ | 2.98 | 1 | 2.98 | 101.26 | <0.01 ** |
| $AB$ | 0.010 | 1 | 0.010 | 0.34 | 0.5780 |
| $AC$ | 0.014 | 1 | 0.014 | 0.49 | 0.5066 |
| $BC$ | $1.6 \times 10^{-3}$ | 1 | $1.6 \times 10^{-3}$ | 0.054 | 0.8222 |
| $A^2$ | 0.13 | 1 | 0.13 | 4.59 | 0.0694 |
| $B^2$ | 1.37 | 1 | 1.37 | 46.70 | <0.01 ** |
| $C^2$ | 0.017 | 1 | 0.017 | 0.59 | 0.4687 |
| Residual | 0.21 | 7 | 0.029 | —— | —— |
| Lack of Fit | 0.15 | 3 | 0.050 | 3.61 | 0.1233 |
| Pure Error | 0.055 | 4 | 0.014 | —— | —— |

Note: $0.01 < p < 0.05$ (significant, *); $0.001 < p < 0.01$ (highly significant, **).

The effects of test factors *A*, *B*, and *C* on tillage depth stability *U* are shown in Figure 17. When the *A* level increased, *U* gradually decreased from 95.3% to 94.1% and the rate of the decrease gradually accelerated in the interval of −0.25~−0.9. The research of Zhang et al. [11] showed that when other factors remained unchanged, the forward speed was proportional to the cutting pitch. The greater the forward speed, the greater the protrusion height at the furrow bottom and the worse the tillage depth stability. The greater the forward speed, the greater the influence on the tillage depth stability. When the *B* level was less than 0, *U* gradually decreased in the interval of 94.9%~95.6%, and the decrease rate gradually tended to 0. When the *B* level was greater than 0, *U* showed an increasing trend in the interval of 94.9%~95.25% and the increase rate gradually increased. The researches of Wu and Yang et al. [1,6] showed that the size and the number of *Cyperus esculentus* tubers increased with the depth, and most of the *Cyperus esculentus* tubers grew to a nearly 115 mm depth. Most of the *Cyperus esculentus* tubers grew close to a depth of 115 mm, which was the test factor tillage depth (*B*) level of 0. A large number of *Cyperus esculentus* tubers squeezed with the soil, which significantly increased the firmness of the soil layer at a 115 mm depth. When the digging depth of the rotary blade was less than 115 mm, the tillage depth stability decreased with the increase in depth. When the digging depth was around 115 mm, the top of the front surface at the maximum rotary radius of the rotary blade collided with the *Cyperus esculentus*–soil agglomeration layer forming displacement deviation, resulting in a dynamic imbalance of the rotary tiller shaft, thus affecting the tillage depth stability. When the digging depth was greater than 115 mm, the cutting edge of the lengthwise blade cut the *Cyperus esculentus*–soil agglomeration layer. Under sliding–cutting, the agglomerate layer broke up and was thrown backward under the action to in front of the scoop surface. With the increase in the *C* level, *U* showed an almost linear growth trend of 94.2%~95.4%. The slope of the curve changed to a small extent and slowly decreased in the interval of 0.6~0.75. The researches of Zhang and Wang et al. [11,15] showed that the shaft speed was inversely proportional to the cutting pitch under the same conditions. The distance between the neighboring trochoids gradually decreased as the cutting pitch decreased. Hence, the protrusion height at the furrow bottom decreased and the tillage depth tended to be stable.

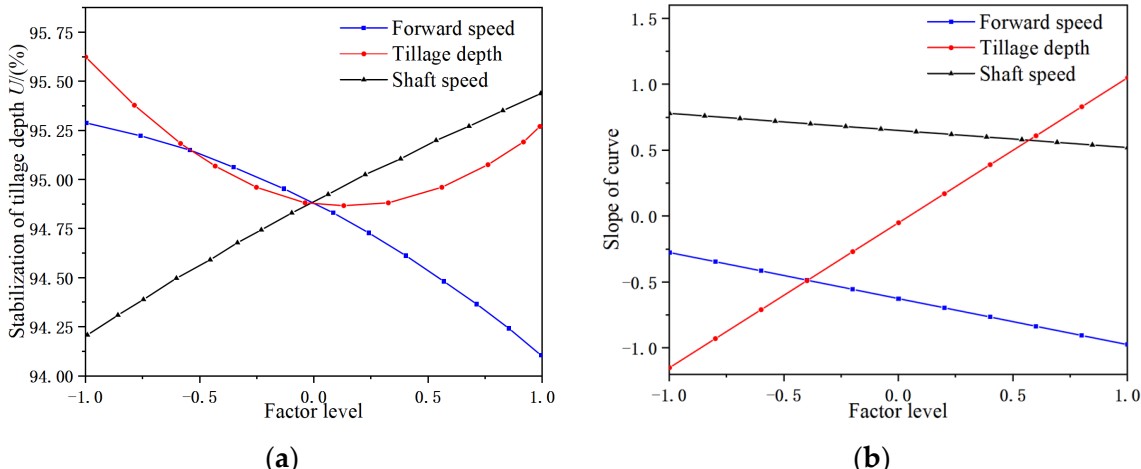

**Figure 17.** Single-factor effect of tillage depth stability. (**a**) Submodel regression curve; (**b**) slope of the factor curve.

The effects of the interaction factors on tillage depth stability *U* were visually analyzed by the response surface plots. As shown in Figure 18, there was no significant change in *U* when the interaction terms (*AB*, *AB*, *BC*) of forward speed, tillage depth, and shaft speed increased. This was consistent with the conclusions in the ANOVA table, that is, that the test factor interaction terms did not have a significant effect on tillage depth stability ($p > 0.05$).

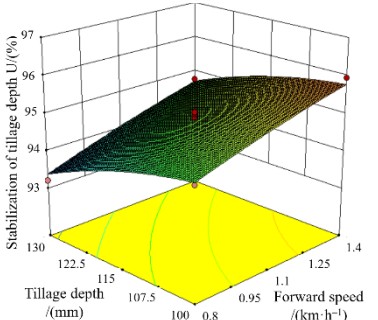 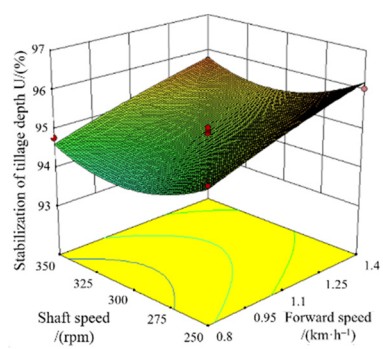 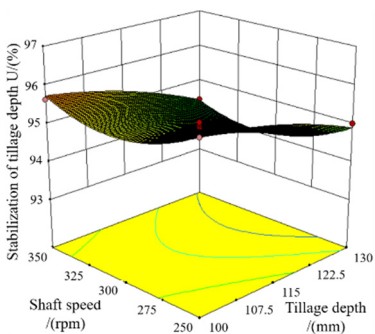

**Figure 18.** Response surface variation relationship influencing factors on the tillage depth stability.

### 3.2.2. Power Consumption of *Cyperus esculentus* Harvesting

Removing non-significant influences, the quadratic polynomial regression equation targeting the power consumption of the rotary tiller was

$$W = 43.37 + 0.59A + 1.19B + 0.82C + 0.48AC - 0.43A^2 + 1.13B^2 \tag{15}$$

In the power consumption ANOVA, as shown in Table 8, the coefficient of determination $R^2 = 0.9657$ indicated that the regression equation model applied to 96.57% of the test data. The effects of the regression model, forward speed ($A$), tillage depth ($B$), rotating shaft speed ($C$), and tillage depth interaction term ($B^2$) were highly significant ($p < 0.01$); machine forward speed and rotating shaft speed interaction term ($AC$) and machine forward speed interaction term ($A^2$) had a significant effect on power consumption ($p < 0.05$). The other test factor interaction terms had no significant effect on power consumption. Neglecting the non-significant factors and comparing the $F$-values, the order of the influence of each factor on the test stacking angle was $B > C > A$.

**Table 8.** Variance analysis of quadratic term model of power consumption.

| Source | Sum of Squares | Freedom | Mean Square | F Value | *p*-Value |
|--------|---------------|---------|-------------|---------|-----------|
| Model | 26.38 | 9 | 2.93 | 21.91 | <0.01 ** |
| $A$ | 2.80 | 1 | 2.80 | 20.90 | <0.01 ** |
| $B$ | 11.26 | 1 | 11.26 | 84.12 | <0.01 ** |
| $C$ | 5.35 | 1 | 5.35 | 39.95 | <0.01 ** |
| $AB$ | $2.0 \times 10^{-3}$ | 1 | $2.0 \times 10^{-3}$ | 0.015 | 0.9056 |
| $AC$ | 0.92 | 1 | 0.92 | 6.89 | 0.0342 * |
| $BC$ | 0.078 | 1 | 0.078 | 0.59 | 0.4690 |
| $A^2$ | 0.77 | 1 | 0.77 | 5.76 | 0.0475 * |
| $B^2$ | 5.40 | 1 | 5.40 | 40.34 | <0.01 ** |
| $C^2$ | 0.034 | 1 | 0.034 | 0.26 | 0.6282 |
| Residual | 0.94 | 7 | 0.13 | —— | —— |
| Lack of Fit | 0.56 | 3 | 0.19 | 1.98 | 0.2596 |
| Pure Error | 0.38 | 4 | 0.094 | —— | —— |

Note: $0.01 < p < 0.05$ (significant, *); $0.001 < p < 0.01$ (highly significant, **).

The effects of test factors $A$, $B$, and $C$ on power consumption $W$ are shown in Figure 19. When the $A$ level increased from $-1$ to 0.3, $U$ gradually increased from 42 to 43.3 and the growth trend gradually slowed down. When the level of $A$ was greater than 0.3, the slope of the $U$ curve tended to 0 and the curve did not change significantly. The research of Zhang et al. [11] showed that the cutting pitch was proportional to the forward speed under the same conditions. When the cutting pitch was small, the cutting resistance increased with the cutting pitch. When the thickness of the soil unit exceeded the critical point, the resistance of the rotary cutter reached the peak and was only related to the rotary blade and soil parameters and was little affected by the forward speed [12,17]. $U$ increased with the increase in $B$ and the growth rate accelerated gradually. However, when the $B$

level increased from −1 to −0.4, *U* varied less from 43 to 43.2 and slightly decreased. The researches of Wang and Hou et al. [5–7] showed that due to the strong tillering ability and developed root system, *Cyperus esculentus* plants were closely connected to the shallow soil. When the tillage depth was small, the cutting resistance of the rotary blade was influenced by the shallow plant leaves and root system, so there was a slight deviation in the effect of the tillage depth on power consumption. Overall, the power consumption is nonlinearly proportional to the tillage depth. The greater the tillage depth, the greater the cutting resistance of the rotary blade and the greater the power consumption [15,19,32,36]. When the *C* level increased, the slope of the *U* curve decreased slightly from 0.9 to 0.8, and overall, the *U* curve showed a linear growth trend. The research of Wang et al. [17] showed that the higher the shaft speed, the greater the power consumption. When the shaft speed was high, the throwing effect of the up-cut rotary blade was enhanced. The soil falling back to the front of the device was reduced and the soil-blocking situation was relieved, so the power consumption showed a slightly decreasing trend in the growth rate when the shaft speed was high.

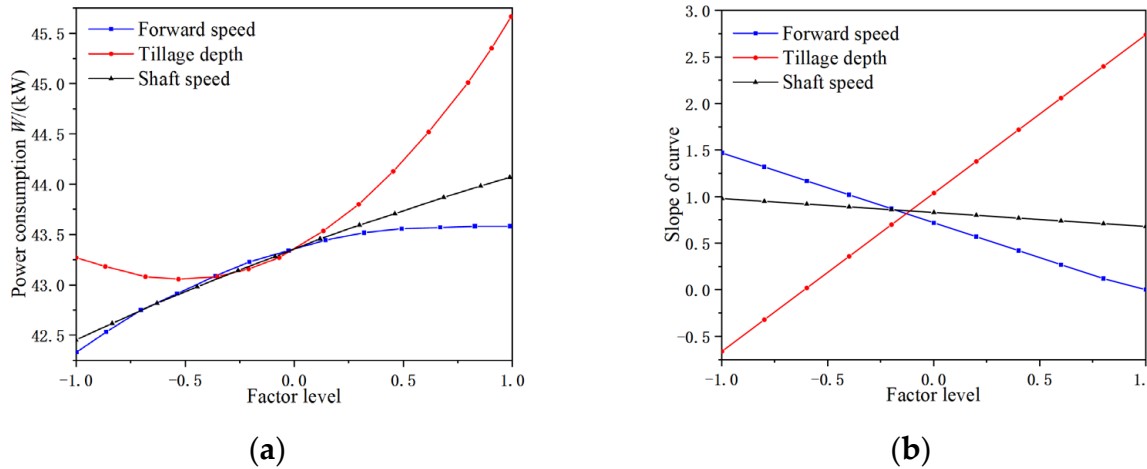

(**a**)             (**b**)

**Figure 19.** Single-factor effect of power consumption. (**a**) Submodel regression curve; (**b**) slope of the factor curve.

The effect of the interaction factors on power consumption *W* is shown in Figure 20. The power consumption of the rotary tiller gradually increased as the interaction terms of the test factors (*AB*, *AC*, *BC*) increased. This is consistent with the conclusion of Zhang et al. [11]. The power consumption *U* increased significantly when the interaction term *AC* increased. This was consistent with the conclusion of the ANOVA table that the effect of the interaction term *AC* on power consumption was significant ($p < 0.05$). Therefore, to reduce the power consumption of the reverse rotary tiller, the test factor chosen should be as small as possible to ensure the harvesting quality of *Cyperus esculentus*.

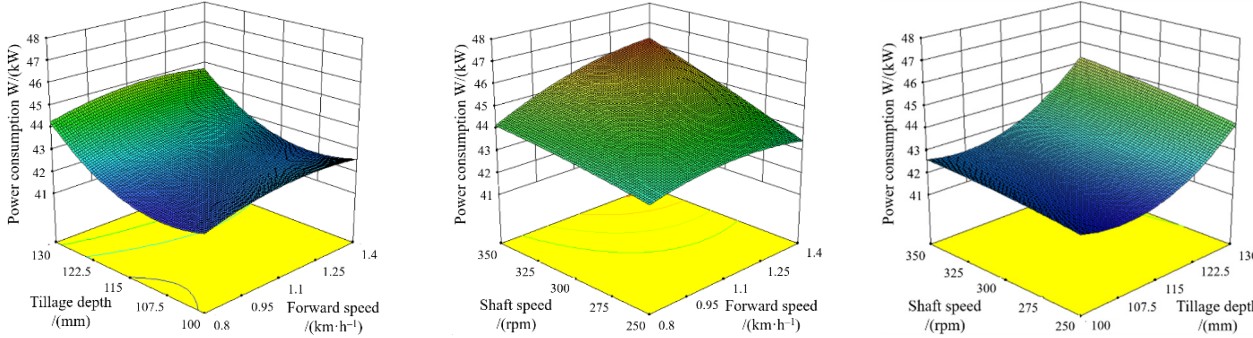

**Figure 20.** Response surface variation relationship of influencing factors on the power consumption.

According to the optimization function of the Design-Expert software, the optimal combination of operating parameters for the reverse rotary tiller for *Cyperus esculentus* harvesting was 1.08 m/s for forward speed, 107.11 mm for tillage depth, and 258.05 r/min for shaft speed. The tillage depth stability was 94.63% and the power consumption was 42.35 kW. Compared with the simulated value of power consumption of the rotary tiller, the practical power consumption increased by 9.10%. The reason is that in the practical operating environment, the sandy loam soil contains a small number of stones and other debris, which will increase the resistance of the rotary tiller in the process of cutting the soil, thus increasing the power consumption [15,19]. In addition, the water content of the sandy loam soil for the established soil model sample was (14.33 ± 1.25)%, whereas the soil water content increased to (15.63 ± 1.56)% due to rainfall in the field test. In the soil's three-phase body (solid-phase skeleton, aqueous solution, and air), the percentage of water increased, the radius of adhesion between the soil particles increased, and the soil cohesion increased, which led to an increase in the soil-breaking resistance of the rotary blade and an increase in the power consumption.

## 4. Conclusions

(1)  In this study, two eccentric arcs with different sliding–cutting performances were used to design a rotary blade edge curve according to the requirements of *Cyperus esculentus* harvesting. By taking equidistant points of the lengthwise blade edge curve, it was calculated that the dynamic sliding–cutting angle was proportional to the pole radius. The dynamic sliding–cutting angle at the tool holder increased rapidly from 35.15° to 46.15°, which reduced plant entanglement and shortened the length of the secondary working area. The dynamic sliding–cutting angle at the tool tip increased slowly from 46.15° to 54.30°, forming a longer primary working area and good plant-crushing performance.

(2)  Based on the discrete element method, a sandy loam soil model and a flexible model of the *Cyperus esculentus* plant were established. The power consumption and plant-crushing performance of the IT245 and IT245P rotary blades were compared by simulation tests. The results showed that the average power consumption of the IT245P rotary blade was reduced by 13.10% and the crushing rate of *Cyperus esculentus* plants was increased by 11.75%.

(3)  The forward speed, tillage depth, and shaft speed were selected as the test factors, and the tillage depth stability and power consumption were selected as the evaluation indexes for the field test. The best combination of operating parameters of the reverse rotary tiller was 1.08 m/s for the forward speed, 107.11 mm for the tillage depth, and 258.05 r/min for the shaft speed. Under this optimal parameter combination, the stability of the tillage depth was 94.63% and the power consumption was 42.35 kW. This study showed that the rotary blade with a double-eccentric circular-edge curve could better complete the plant-crushing and power consumption reduction operations and meet the requirements for the harvesting of *Cyperus esculentus*.

**Author Contributions:** Conceptualization, H.Z. and X.H.; methodology, H.Z.; software, H.Z. and H.W.; validation, H.Z., D.W. and Z.Z.; formal analysis, H.Z. and Y.T.; investigation, H.Z.; resources, H.Z.; data curation, C.L.; writing-original draft preparation, H.Z.; writing-review and editing, S.S.; visualization, H.Z.; supervision, H.Z.; project administration, X.H.; funding acquisition, X.H. All authors have read and agreed to the published version of the manuscript.

**Funding:** This research was funded by the Autonomous Region Science and Technology Support Project Plan (Grant NO.2020E02112) and Major Science and Technology Projects in Henan Province (Grant NO.211100110100).

**Institutional Review Board Statement:** Not applicable.

**Informed Consent Statement:** Not applicable.

**Data Availability Statement:** Not applicable.

**Conflicts of Interest:** The authors declare no conflict of interest.

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
