# Peer review of "Evaluation of Soil-Cutting and Plant-Crushing Performance of Rotary Blades with Double-Eccentric Circular-Edge Curve for Harvesting Cyperus esculentus"

_agriculture, doi:10.3390/agriculture12060862_

Round 1
Reviewer 1 Report
1. Typos are present in the text (and also in the figures), please make corrections.
2. English revision is necessary, at times, it is difficult to understand.
3. In page 8, the authors refer to figure 10, but it should be checked because the explanation relates to figure 9.
4. Not all figures and tables are discussed in the text, and they should be.
5. I advise the authors to extend the literature research beyond their own country, to an international level, thus strengthening the quality of the research.
Reviewer 2 Report
An interesting and well developed article. The authors have combined theoretical and practical knowledge well. They carried out the theoretical selection of the geometry of the working tooth blade of the machine for liquidation of the plantation of the plant Cyperus esculentus, and then, after making the developed tools, they tested them in the laboratory and in the field conditions. The results of the tests confirmed the assumptions made and the theoretical calculations. Based on the results of their analysis and research obtained from the theoretical and practical parts, they made correct conclusions.
The paper, although interesting, will not arouse much interest in readers because it deals with tools misrepresented for the elimination of plantations of a not very popular plant. The authors should look for the possibility of wider application of the developed working tools. The plant Cyperus esculentus is cultivated not only in the authors' homeland but also in the wine-growing parts of the world. In the bibliography, unfortunately, there are practically no publications by authors from outside China. This should be completed in the descriptions in the introduction and in the bibliography, as the authors should do a full international literature review. Many different machines are used for plantation eradication in the world and such a paragraph with relevant sources should also be included in the introduction. In it, the machines and tools used so far in China and the transition to the reason, already described in the text of the paper, for the development of a new shape of knife that shreds the stems and roots of the plant Cyperus esculentus.
Other errors and omissions were also noted in the text:
Line 216: The authors write:" according to the reference literature", but do not provide any source works on which they make this claim. This should be supplemented.
Line 217: The authors write about four models of soil structure, but in line 218 and Figure 7 they show three. The text should be corrected or Figure 7 should be supplemented with the missing model.
Line 228: It would be helpful to add a photo of the study object, the field with the Cyperus esculentus plants to be destroyed.
Line 276: The caption of Figure 10 should be on the same page as the figure, but is on the next page. This should be corrected.
Line 331: The textbook mentioned should be added to the bibliography list since its title appears in the text.
Line 395: Table 6 should be placed entirely on one page, and if it must be divisible, it should not be done in the ways presented in the article. You cannot leave the headings on one page and put the compactness of the table on another page because it is difficult to interpret.
Places for improvement are marked in the text of the article.
It would be advisable to insert a line break before each figure for better readability of the paper as it is before tables.

Reviewer 3 Report
- Please include some of the patents related to improving tillage efficiency and reducing energy intensity.
- It is necessary to present the errors of measuring instruments when conducting studies of the energy costs of the developed working body for tillage.
- When conducting theoretical studies to justify the developed working body, it is necessary to take into account the physical and mechanical properties of the soil.
- It is necessary to provide the values of soil hardness and density during research.
